# Quantification of Very Low Concentrations of Colloids with Light Scattering Applied to Micro(Nano)Plastics in Seawater

**Gireeshkumar Balakrishnan \***, **Fabienne Lagarde, Christophe Chassenieux and Taco Nicolai**

Institut des Molécules et Matériaux du Mans, UMR CNRS 6283, Le Mans Université, Avenue Olivier Messiaen, CEDEX 9, 72085 Le Mans, France; fabienne.lagarde@univ-lemans.fr (F.L.); christophe.chassenieux@univ-lemans.fr (C.C.); taco.nicolai@univ-lemans.fr (T.N.)
* Correspondence: gireeshkumar.balakrishnan_nair@univ-lemans.fr

**Abstract:** The detection and quantification of micro(nano)plastics in the marine environment are essential requirements to understand the full impacts of plastic pollution on the ecosystem and human health. Here, static light scattering (SLS) and dynamic (DLS) light scattering techniques are assessed for their capacity to detect colloidal particles with diameters between d = 0.1 and 0.8 μm at very low concentrations in seawater. The detection limit of the apparatus was determined using model monodisperse spherical polystyrene latex particles with diameters of 0.2 μm and 0.5 μm. It is shown that the concentration and size of colloids can be determined down to about $10^{-6}$ g/L. Light scattering measurements on seawater obtained from different locations in Western Europe show that colloidal particles were detected with DLS in seawater filtered through 0.8 μm pore size filters. The concentration of these particles was not higher than 1 μg/L, with an average diameter of about 0.6 μm. We stress that these particles are not necessarily plastic. No particles were detected after filtration through 0.45 μm pore size filters.

**Keywords:** nanoplastics; microplastics; seawater; colloids; static light scattering; dynamic light scattering

## 1. Introduction

The fate of plastics that end up in the sea is currently attracting much attention [1]. When discharged in the environment, plastics undergo mechanical (erosion, abrasion), chemical (photo-oxidation under UV radiation, hydrolysis), and biological (degradation by microorganisms) actions [1–6], which leads to aging and fragmentation of macroplastics into microplastics, defined as plastic particles smaller than 5 mm [7]. Microplastics were found to be ubiquitous in the environment, in particular on the surface of the oceans. One important issue that has recently emerged is whether microplastics continue to fragment into colloidal particles with a diameter d < 1 μm that are often called nanoplastics [8–10]. However, some authors consider plastic particles to be nanoplastics provided their diameter is less than 100 nm or use the expression micro(nano)plastics [11,12]. For simplicity, we will call the particles with d < 1 μm that were detected in this study nanoplastics.

Gigault et al. [8] investigated the release of nanoplastics under UV light from weathered polyethylene and polypropylene fragments sampled from the environment. They observed that nanoplastics with a broad range of sizes were produced over a period of weeks. More recently, Ter Halle et al. [13] investigated seawater collected near the surface of the North Atlantic subtropical gyre. The seawater was filtered through 1.2 μm pore size filters, and the filtrate was inspected for the presence of nanoplastics with dynamic light scattering. Particles with diameters between 1 nm and 1 μm were detected in seawater that was concentrated by a factor of 200. It was suggested that these particles were mostly nanoplastics formed by the degradation of microplastics, but the authors did not provide an estimate of the concentration of nanoplastics in the seawater.

The presence of such small particles raises questions about their environmental concentration and their potential accumulation in the trophic chain. Indeed, due to their small size and specific properties, nanoplastics can be ingested by a very large range of aquatic organisms and can interact with membranes and cells [4,12]. Nanoplastics dispersed in the seawater could be part of the "lost plastic" that has been dumped in the sea but is no longer observed at the surface [5,14,15]. Detection and quantification of nanoplastics in all aquatic compartments are, therefore, urgent needs. A major difficulty is that even though the total amount of plastic in the sea is huge, the concentration of nanoplastics in seawater is still expected to be very low.

Recently, various techniques have been developed to detect nanoplastics in the natural environment [16]. One method involves fluorescently-labeling the nanoplastics [17,18] in order to detect them with a microscope. This method is easy to perform and can be used to measure the average size and shape of the particles. However, in most cases, the binding between the dye and plastic is not covalent, and hence leaching of the dye occurs [19]. One common dye used for labeling plastic is Nile Red [20], which generates crystals in water that lead to false identification as plastic particles. Another method that has been widely used in the literature is gas chromatography coupled with mass spectroscopy after pyrolysis [13]. The advantage of this method is that the limit of detection of plastics is down to a few µg/L. The main limitation of this method is the interference from natural organic matter in the aquatic environment [21]. Another promising strategy is the use of AF4 combined with multi-angle light scattering for the detection of nanoplastics in food matrices such as fish [22]. It was reported that PS nanoplastics down to a concentration of 52 µg/g of fish could be detected, but no detection was possible for PE particles. This shows that the method developed for PS cannot be applied to other types of plastics. Yet another strategy is the combination of optical tweezers with Raman spectroscopy for the detection of plastic particles with sizes between 50 nm and 20 µm [23]. This method can be used to detect and isolate nanoplastics among natural organic and mineral particles in the aquatic environment, but their quantification remains difficult.

Static and dynamic light scattering techniques have the potential to yield both the average size and the concentration of colloidal particles, even if they are present at very low concentrations. The aim of the investigation reported here was twofold. First, we critically assess the potential of these light scattering techniques to quantify the concentration and size distribution of model nanoplastics in the form of polystyrene latex particles with d < 0.8 µm dispersed in water. Then we will discuss light scattering measurements on seawater sampled at different places near the coast of Western Europe. We show that the concentration of colloids with 0.2 < d < 0.45 µm is less than 1 µg/L and cannot be characterized by light scattering. The concentration of colloids with 0.45 < d < 0.8 µm is approximately 1 µg/L and can be characterized if care is taken. Of course, colloids that are detected in seawater do not necessarily consist of plastic, as mineral colloids are expected to be present. We will mention in the Discussion section how the light scattering results depend on the type of material.

## 2. Materials and Methods

### 2.1. Materials

Model polystyrene latex particles with diameters of d = 0.2 µm (Thermofisher, catalog number: 5020A) and d = 0.5 µm (Polysciences, catalog number: 15700) were used as received. The density of the polystyrene particles was given by the provider as 1.05 g/mL. The solid content of the latex suspensions was 10 wt% and 2.5 wt% for d = 0.2 µm and d = 0.5 µm, respectively. Particle suspensions were prepared by dilution with salt-free Milli-Q water at different concentrations: C = $10^{-3}$, $10^{-4}$, $10^{-5}$, and $10^{-6}$ g/L. These standard PS latex particle surfaces are modified with a carboxylate group, which is introduced during their synthesis. The negative surface charge on the particles makes them easily dispersible in Millipore water and also makes them stable against aggregation. Neither agglomeration nor sticking of the particles to the container walls were observed.

Samples of seawater (about 2 L) were taken by hand near the surface at different locations near the coast of France, Spain, and the Netherlands: Roscoff (48°43′35.9″ N 3°58′57.0″ W), Challans (47°03′49.4″ N 2°00′41.2″ W), Toulon (43°07′15.8″ N 5°55′28.1″ E), Lanzarote (28°57′23.8″ N 13°33′16.6″ W), and Wassenaar (52°08′55.5″ N 4°19′46.6″ E) in plastic or glass bottles that had been extensively rinsed with the same seawater. At one of these locations (Toulon), seawater was taken both near the surface and, by divers, at a depth of 10 m. The samples were filtered through Acrodisc nylon membrane filters with pore sizes of 0.45 μm or MF-Millipore cellulose ester membrane filters with pore sizes of 0.8 μm. No colloidal particles could be detected by light scattering (DLS) in Milli-Q water filtered through 0.45 μm pore size filters. When colloid free Milli-Q water was filtered through the 0.8 μm filters, colloidal particles were detected with DLS, showing that these filters released particles. Therefore, it was necessary to wash the filters by filtering about 50 mL of Milli-Q water until the release of particles was no longer detected by light scattering. A number of other commercial filters were tested (Acrodisc glass membrane filters (1 μm), Whatman poly (ether sulphone) membrane filters (0.8 μm), and Whatman glass fiber filters (1.5 μm)), but they released more particles and were therefore discarded. We tested the retention of particles smaller than the pore size by comparing the scattering intensity of latex particles before and after filtration and found it to be negligible. In addition, we did not find that the scattering intensity decreased further if filtered solutions were filtered a second time.

## 2.2. Light Scattering

The theory of static and dynamic light scattering is briefly reviewed here. For more details, see Refs. [24–29]. In static light scattering, the average scattering intensity of the scattering objects is measured for a given interval of time. This average scattering intensity is often expressed in terms of the Rayleigh ratio ($R_\theta$) which is calculated as the average excess scattering intensity of the samples over that of the solvent normalized by the scattering of a standard, for which we used toluene with Rayleigh factor of $Rref = 1.35 \times 10^{-5}$ cm$^{-1}$. Using the Rayleigh–Gans approximation, one can relate $R_\theta$ to the average molecular weight ($M_w$) and the structure factor $S(q)$ of the scattering objects by the following equation.

$$R_\theta = K \cdot C \cdot M_w \cdot S(q) \tag{1}$$

where $K$ ($K = 4\pi^2 n^2 \left(\frac{\delta n}{\delta c}\right)^2 / \lambda^4 N_a$) is an optical constant that depends on the refractive index increment ($\delta n / \delta c$) and the wavelength of the light ($\lambda$). $C$ is the concentration of the scattering objects and $N_a$ is Avogadro's number. $S(q)$ describes the dependence of the scattered intensity on the scattering wave vector $q$, which itself is a function of the angle of observation $\theta$ ( $q = \frac{4\pi n}{\lambda} \sin\left(\frac{\theta}{2}\right)$).

For mono-disperse spherical particles of diameter $d$:

$$S(q) = \left[3(\sin\left(\frac{qd}{2}\right) - \frac{qd}{2}\cos(\frac{qd}{2})) / \left(\frac{qd}{2}\right)^3\right]^2 \tag{2}$$

For particles of any shape, the initial $q$-dependence of $S(q)$ can be expressed as a series expansion in terms of the z-average radius of gyration ($R_g$):

$$S(q) = \left[1 + \frac{q^2 R_g{}^2}{3} + \ldots\right]^{-1} \quad q \times R_g < 1 \tag{3}$$

With DLS, one determines the correlation between the intensity at a given time with that at a delay time *(t)* later. The average over many starting times yields the normalized au-

tocorrelation function of the scattered light intensity $g_2(t)$. $g_2(t)$ is related to the normalized electric field autocorrelation function $(g_1(t))$ through the so-called Siegert relation:

$$g_2(t) - 1 = \beta \left[ g_1(t)^2 + \frac{<\delta N(0)\delta N(t)>}{N^2} \right] \tag{4}$$

The pre-factor $\beta$ is smaller than unity and depends on the optical set-up. The second term in Equation (4) reflects the fluctuation in the number of particles ($N$) that are present in the scattering volume. The $g_1(t)$ obtained from dynamic light scattering measurements, could be described by a monomodal relaxation time distribution ($A(\tau)$), which was determined by fitting the correlation functions to a general exponential function:

$$g_1(t) = \int A(\tau) \exp\left(\frac{-t}{\tau}\right) d\tau \tag{5}$$

with

$$A(\tau) = H\tau^p exp\left(-\frac{\tau}{\tau_{gex}}\right)^s \tag{6}$$

where $H$ is a normalization constant, $\tau_{gex}$ is a characteristic relaxation time and $p$ and $s$ are parameters that allow different shapes of the distribution. The average relaxation rate ($\Gamma = <1/\tau>$) is related to the z-average diffusion coefficient ($D$) of the particles in the suspension according to:

$$\Gamma = Dq^2 \tag{7}$$

From the diffusion coefficient, the z-average hydrodynamic diameter ($d_h$) of the particles was calculated using the Stokes–Einstein relation:

$$D = \frac{kT}{3\pi\eta d_h} \tag{8}$$

where $k$ is the Boltzman constant and $\eta$ the viscosity of the solvent.

Static and dynamic light scattering measurements were conducted using a commercial apparatus ALV/CGS3 (ALV, Langen, Germany). The light source was a He-Ne laser with wavelength $\lambda = 632$ nm. The temperature was controlled by a thermostat bath to $20 \pm 0.2$ °C. Measurements were made at angles of observation ($\theta$) between 13 and 150 degrees, which correspond to scattering vectors $q$ (ranging from $3.0 \times 10^6$ up to $2.5 \times 10^7$ m$^{-1}$. Intensity autocorrelation functions were obtained using a digital multi-tau correlator.

## 3. Results

### 3.1. Model Particles

We tested the limitations for static and dynamic light scattering measurements of the equipment used in this study with monodisperse polystyrene latex particles with $d = 0.2$ μm and $d = 0.5$ μm. Figure 1 shows $R_\theta$ as a function of q for the aqueous latex suspensions at C = $10^{-3}$, $10^{-4}$, $10^{-5}$, and $10^{-6}$ g/L. For comparison, we also show the results for pure Milli-Q water. The dashed lines in Figure 1 represent fits to Equations (1) and (2) with d = 0.2 μm and d = 0.50 μm, whereas the solid lines represent fits to the Mie theory [26]. The Rayleigh-Gans approximation (Equation (1)) gave similar results for the smaller particles, but the Mie theory described the experimental results better for the larger particles at higher q-values. The experimental value of $R_\theta$ for pure water found in this investigation is in good agreement with the value reported in the literature [30] and is shown for comparison in Figure 1a.

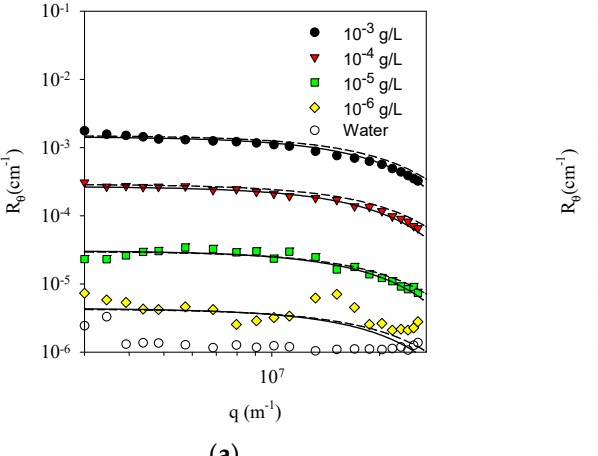
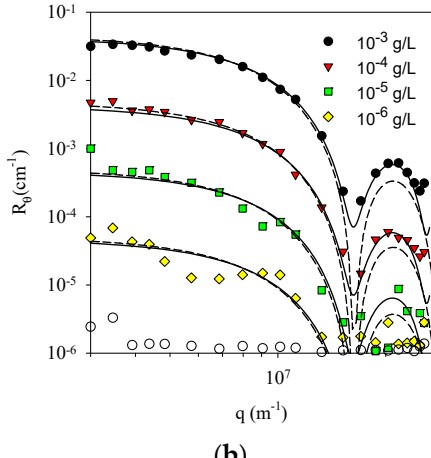

(a)  (b)

**Figure 1.** Dependence of the Rayleigh ratio on the scattering wave vector for suspensions of latex particles with d = 0.2 μm (**a**) or d = 0.5 μm (**b**) at different concentrations. The dashed and solid lines represent fits to the theory assuming the Rayleigh-Gans approximation Equation (1) and the Mie theory, respectively. The symbols and colors are the same in (**a**,**b**).

It is important to realize that, as a consequence of the steep decrease of $R_\theta$ with increasing q for $q > d^{-1}$, suspensions of the smaller particles actually scatter more light for $q > 2 \times 10^7$ m$^{-1}$ ($\theta > 70°$) than those of the larger particles at the same concentration, see Figure 1. The scattering intensity of the smaller latex suspension was much larger than that of water down to $C = 10^{-5}$ g/L over the whole accessible q-range. However, $R_\theta$ of the suspension of the larger latex particles at $C = 10^{-5}$ g/L approached that of water at the highest q-values. As a general feature, the scattering intensity by suspensions of homogeneous spherical particles at a fixed mass concentration increases with increasing size for q×d << 1, but decreases for $q \times d > 1$ as can be clearly seen from Figure 1. At a given value of q and C, $R_\theta$ is largest for particles with $d \approx 2\pi/q$. It is, therefore, necessary to do light scattering measurements at small q-values if very low concentrations of large particles are investigated. This is illustrated here for particles with d = 0.5 μm for which the scattering intensity is close to that of water at $C \leq 10^{-5}$ g/L if $q > 2 \times 10^7$ m$^{-1}$, i.e., if $\theta > 70°$, but at smaller angles they still scatter orders of magnitude more light than water even at $C = 10^{-6}$ g/L, see Figure 1b.

As was mentioned above, if the number of particles in the scattering volume (N) is not large, one needs to consider the fluctuation of N in time due to the diffusion of particles in and out of the scattering volume (see Equation (4)). This effect can be clearly seen from the time dependence of the intensity at different concentrations. Figure 2 shows examples of $R_\theta$ as a function of time for the larger latex particles at $C = 10^{-4}$, $10^{-5}$, and $10^{-6}$ g/L at $q = 3.9 \times 10^6$ m$^{-1}$ ($\theta = 17°$). For each solution, three measurements were conducted for a duration of 15 min. The average value of $R_\theta$ decreases in proportion to the concentration, but slow fluctuations became significant for $C = 10^{-5}$ g/L and were more important for $C = 10^{-6}$ g/L. The scattering volume of the apparatus used here was approximately 0.3 mm$^3$. The average number of particles in this volume was 45 at $C = 10^{-5}$ g/L and less than 5 at $C = 10^{-6}$ g/L. It takes the latex particles about $10^3$ s to diffuse 0.1 mm, which explains why the fluctuations in $R_\theta$ are very slow. As a consequence, one needs to average over very long time periods to obtain accurate averages.

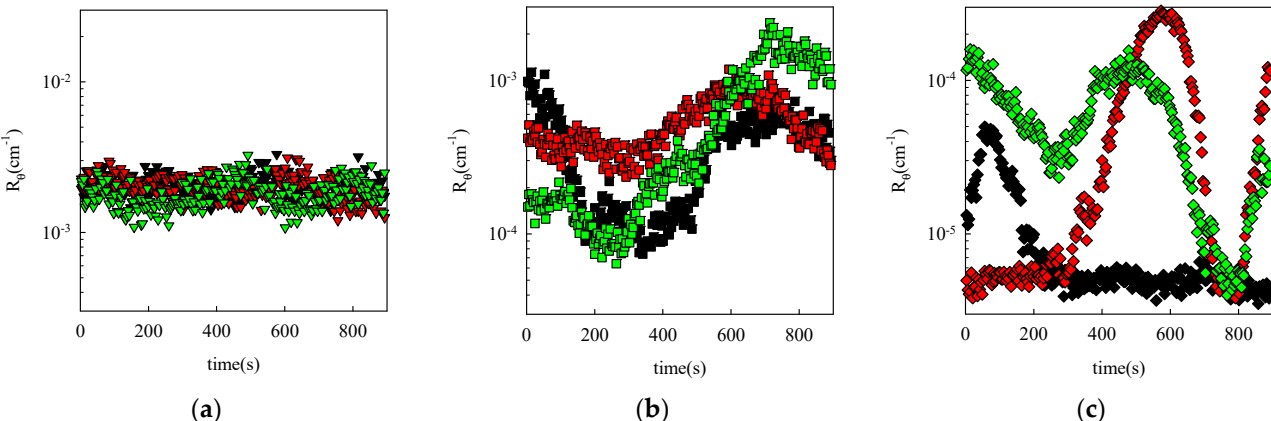

**Figure 2.** Time dependence of $R_\theta$ for suspensions of latex particles (d = 0.5 µm) at C = $10^{-4}$ ((**a**), triangles), $10^{-5}$ ((**b**), squares), and $10^{-6}$ ((**c**), diamonds) g/L at q = 3.9 × $10^6$ m$^{-1}$ (θ = 17°). The red, green and black colors represent three independent measurements for each concentration.

Dynamic light scattering measurements could not be conducted reliably for suspensions of the larger latex particles at C = $10^{-6}$ g/L because the average number of particles in the scattering volume was too low. Figure 3a shows intensity autocorrelation functions obtained at different scattering vectors for latex particles with d = 0.5 µm at C = $10^{-5}$ g/L. The correlation functions were analyzed using Equation (5), assuming a log-normal size distribution. The solid lines in Figure 3a represent the fit results, and the corresponding size distributions are shown in Figure 3b. The q-dependence of the z-average hydrodynamic diameter is shown as an inset in Figure 3b. Even at this low concentration, the $d_h$ values found with DLS were within 20% of the nominal value at low q-values and within 40% at high q-values. The lower precision at higher q-values was caused by the low scattering intensity; see Figure 1. Notice that the correlation functions shown in Figure 3a did not all reach zero, which was due to the slow fluctuation of the number of particles in the scattering volume discussed above that causes an additional slow relaxation time at very low particle concentrations. This problem was greatly exacerbated at C = $10^{-6}$ g/L and is the reason why no reliable DLS results could be obtained for that system.

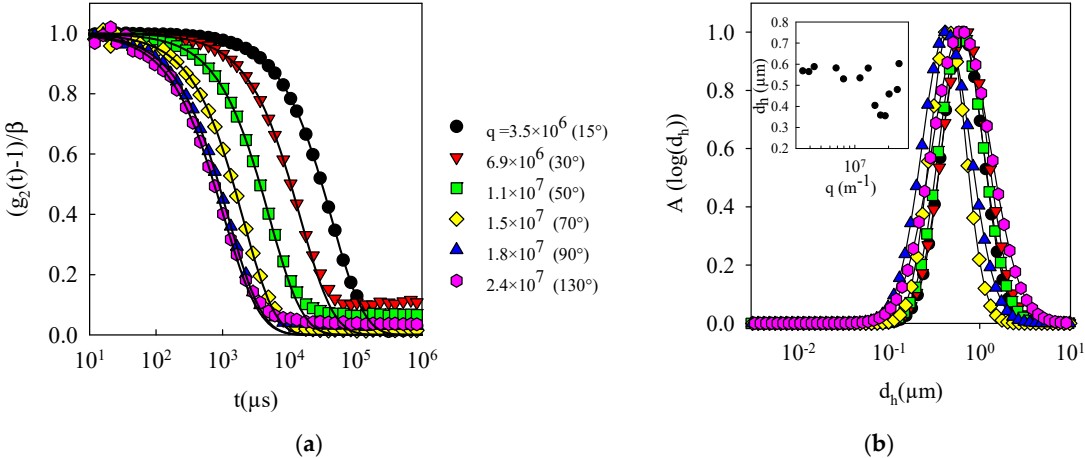

**Figure 3.** (**a**) Normalized intensity autocorrelation functions at different scattering wave vectors (q(m$^{-1}$)) obtained for suspensions of latex particles with d = 0.5 µm at C = $10^{-5}$ g/L. The corresponding scattering angles are indicated in brackets. The solid lines represent the fit results to Equation (5). (**b**) Distributions of the hydrodynamic diameter corresponding to the fit results shown in (**a**). The inset shows the z-average hydrodynamic diameter as a function of q. Please note that the symbols and colors in (**b**) is same as (**a**).

### 3.2. Colloidal Particles in Seawater

The capacity to detect and characterize colloidal nanoplastics in seawater was tested by investigating samples of seawater. The seawater was filtered through 0.8 μm or 0.45 μm pore size filters in order to assess the presence of particles smaller than 0.8 μm and smaller than 0.45 μm separately. This is necessary because the presence of a small number of large particles can hide the light scattering signal from small particles. Figure 4 shows the q-dependence of $R_\theta$ in comparison with that of Milli-Q water to which sea salt was added at the concentration found in the sea. As expected, adding sea salt caused a small increase in the scattering intensity with respect to pure water, as shown in Figure 1 [30].

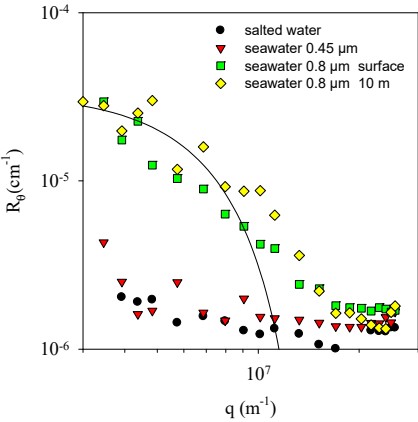

**Figure 4.** Dependence of the Rayleigh ratio on the scattering wave vector for seawater filtered using different pore sizes. For comparison, the results for Milli-Q water with added sea salt are also shown. The solid lines represent a fit to Equations (1) and (2) with d = 0.6 μm. Black circle for salted Milli-Q water, triangle down for filtered seawater (0.45 μm filter), and a green square and yellow diamond for filtered seawater (0.8 μm filter) collected near the surface and at a depth of 10 m, respectively.

The amount of light scattered by seawater filtered through 0.45 μm pores was within the experimental error, the same as for Milli-Q water with the right amount of sea salt over the whole q-range. Considering results obtained on model latex particles discussed in the previous section, such a result would be obtained for suspensions of latex particles with d = 0.2 μm only if C < 1 μg/L and for particles with d = 0.5 μm only if C < 0.1 μg/L. The implication is that the concentration of particles in this sample of seawater with d between 0.2 and 0.45 μm was less than 1 μg/L, where we assume that the refractive index increment and the density of the particles are close to those of polystyrene, which is the case for most types of nanoplastics. Of course, this does not exclude the possibility that smaller particles are present in higher concentrations. For instance, the excess scattering intensity of very dilute suspensions of particles with d = 20 nm is a thousand times less than for d = 0.2 μm at the same mass concentration.

The time-averaged value of $R_\theta$ of seawater filtered through 0.8 μm pores was much larger and decreased strongly with increasing q, which shows that it was dominated by the scattering from large particles. The solid line through the data represents a fit to monodisperse spherical particles with d = 0.6 μm. The deviation at q > $10^7$ m$^{-1}$ means that the particles are not monodisperse spheres, as might be expected. Comparison with the results obtained for the latex particles with d = 0.5 μm shows that the concentration of particles in the seawater with d between 0.45 and 0.8 μm was less than 1 μg/L, assuming that their refractive index and density are close to those of polystyrene. Results obtained with seawater sampled at different locations and at different depths were similar.

Figure 5 shows the scattering intensity as a function of time for seawater filtered through 0.8 μm pores taken at the surface and at a depth of 10 m. $R_\theta$ fluctuated slowly with time, indicating that the number of particles in the scattering volume was not large, as

was discussed above. However, the average value of $R_\theta$ was the same at the two different depths. Results obtained with seawater samples taken at other locations were similar.

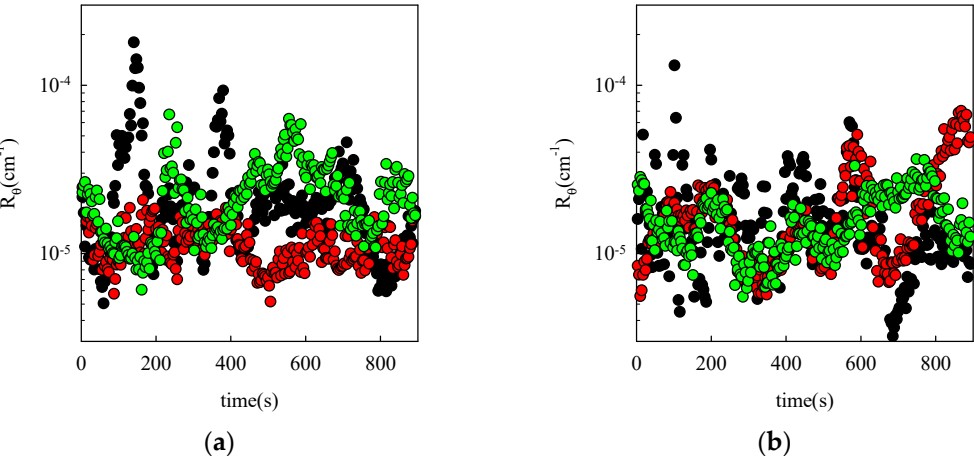

**Figure 5.** Time dependence of $R_\theta$ at $q = 3.9 \times 10^6$ m$^{-1}$ ($\theta = 17°$) for suspensions of seawater filtered through 0.8 µm pore size filters taken at the surface (**a**) and 10 m depth (**b**). The red, green and black colors represent three independent measurements in each case.

Autocorrelation functions of seawater filtered with a pore size of 0.45 µm did not show significant relaxation with $(g_2(t) - 1) \approx 0$ for $t > 1$ µs. This is expected as the scattering by seawater is caused by density fluctuations and the diffusion of ions, which relax on timescales shorter than 1 µs. Figure 6a shows examples of normalized intensity autocorrelation functions obtained at different scattering angles for seawater filtered with a pore size of 0.8 µm. Notice that results obtained at higher scattering angles were not trustworthy because the scattering intensity was close to that of seawater; see Figure 4. The correlation functions show a well-defined fast decay followed by an ill-defined slow decay. The fast decay is due to the diffusion of particles, whereas the slow decay is caused by fluctuations in the number of particles in the scattering volume.

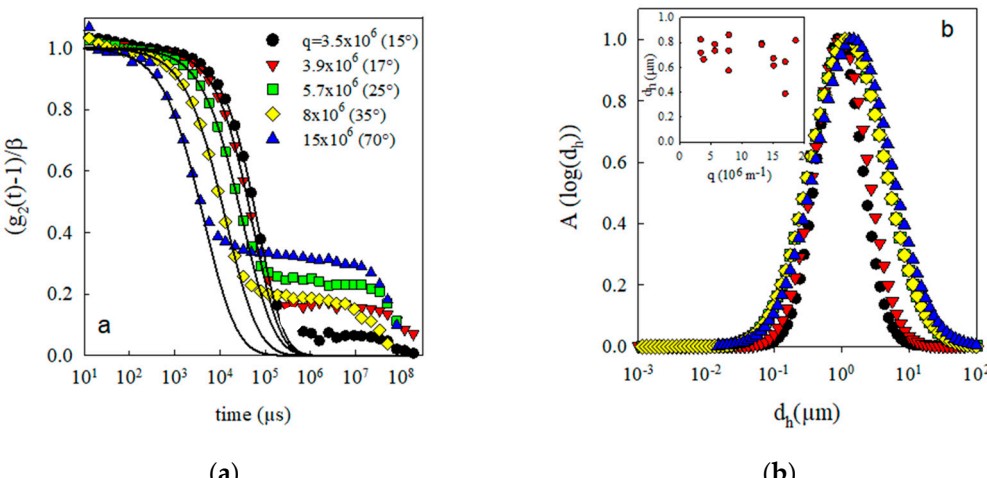

**Figure 6.** (**a**) Normalized intensity autocorrelation functions at different scattering wave vectors obtained for seawater filtered through 0.8 µm pores. The solid lines represent the fit results to Equation (5). (**b**) Distributions of the hydrodynamic diameter corresponding to the fit results shown in Figure 6a. The inset shows the z-average hydrodynamic diameter as a function of q. Please note that the symbols and colors in (**b**) is same as (**a**).

The fast decay was analyzed in terms of a relaxation time distribution that was converted into a distribution of $d_h$. The fit results are shown as solid lines in Figure 6a,

and the corresponding size distributions are shown in Figure 6b. In most cases, z-average hydrodynamic diameters between 0.6 and 0.8 μm were obtained, consistent with the diameter obtained from fitting the structure factor (0.6 μm). The relatively weak dependence of $d_h$ on q implies that the particles were roughly spherical and not very polydisperse, but the structure factor shows that they are not perfect monodisperse spheres either. Notice that the size distribution extends to sizes larger than the pore size. The reason is that the analysis method gives a distribution of sizes even if the particles are monodisperse. This can be clearly seen from the results on monodisperse latex particles shown in the previous section. The average diameter does, however, correspond to the true average diameter. A second reason why the distribution extends to larger values than the nominal pore size is that the 0.8 μm filters contain a distribution of pore sizes and may therefore allow some larger particles to pass.

## 4. Discussion

It was demonstrated here that it is possible to quantitatively characterize colloidal particles in aqueous suspension with static and dynamic light scattering as long as they scatter significantly more than water and the scattering volume contains at least a few tens of particles. These limitations depend on the size, shape, polydispersity, and refractive index increment of the colloids. Monodisperse spherical latex particles with d = 0.2 μm could be reliably characterized by static light scattering down to C = 10 μg/L. Latex particles with d = 0.5 μm could be characterized by static light scattering measurements down to C = 1 μg/L, but only down to C = 10 μg/L by DLS. Of course, there is no sharp boundary between concentrations that can and cannot be characterized by light scattering techniques. It is simply the case that the results become progressively less reliable when the concentration decreases.

For samples of seawater, we found that the concentration of colloids with diameters between 0.2 and 0.8 μm, was not more than 1 μg/L, assuming that they have the same refractive index increment and density as polystyrene. This concentration was barely sufficient for quantitative characterization by light scattering. The scattering of seawater filtered through 0.45 μm pores was within the experimental error, the same as that of salted water. This means that the intensity detected for seawater filtered through 0.8 μm pore size filters is due to scattering by colloids with diameters between 0.45 μm and approximately 0.8 μm.

If we consider that the amount of "lost plastic", which is estimated at about $10^{14}$ g [5], is distributed equally in the form of colloids in the oceans, which have a total volume of about $10^{21}$ L, the expected concentration of nanoplastics is at most 0.1 μg/L, which was shown here to be below the limit of detection by light scattering techniques. We did not observe major differences in the amount of larger colloids in the seawater samples taken at different locations. However, these samples were all taken near the coast of Europe and may therefore not be representative of the global average concentration. On the other hand, Erikson et al. [5] found that the distribution of microplastics (between 0.33 and 1 mm) in the North Atlantic was within a factor of 2 the same as in the other oceans. In addition, some of the samples presented here were taken in the Mediterranean Sea, which is known to be a hotspot for plastic pollution [31]. More measurements of the concentration of colloids at different locations and depths are needed to determine their actual distribution in the oceans.

The formation of larger microplastics due to plastic fragmentation in the marine environment is well established, and its adverse effects on aquatic organisms are a serious concern. However, we still lack data to support the formation of nanoplastics with d < 1 μm in the marine environment. If we assume that these particles are formed, then the question is whether they are stable in the marine environment. It is most likely that the particles are charged when their size is <1 μm and hence the presence of salt will induce aggregation due to screening of the surface charge of the particles [32,33]. Interestingly, the presence of organic matter does not prevent the aggregation of PS nanoplastics in seawater (see Refs. [21,22]). In addition, it was reported that in the presence of salt, UV irradia-

tion induces the aggregation of nanoplastics [34,35]. Hence, most likely, in the marine environment, nanoplastics are in aggregated form.

Unfortunately, DLS cannot inform us about the chemical composition of the detected particles. It is therefore not possible to determine whether the detected colloids are actually nanoplastics. One also needs to consider that there are many natural sources of colloids in the ocean [36–38]. Kioke et al. [39] reported the presence of large numbers ($10^7$ per ml) of submicron detrital particles in the open ocean. They found that 95% of those particles are between 0.32 and 0.6 μm and these particles were produced by the activity of small flagellates. These particles account for almost 10% of the total dissolved organic materials in the ocean.

Interestingly, it has been reported that colloidal particles form spontaneously within hours or days in seawater that was filtered through 0.45 μm or 0.22 μm pore size filters [40,41], which was attributed either to the association of dissolved organic matter into polymer gel particles [40] or to the spontaneous formation of mineral-organic particles [41]. We have tested whether colloids were formed in the filtered seawater samples studied here with time for up to two weeks, but we did not observe that the scattering intensity increased in any of the filtered seawater samples that were collected for this study. A possible explanation is that the glassware used in the studies reported in the literature slowly released colloidal particles. We have ourselves noted this in the past.

As mentioned in the introduction section, Ter Halle et al. [13] address the issue of the identification of nanoplastic particles from the large source of colloidal particles present in the ocean. They used different DLS equipment that allowed measurements only at a single high scattering angle ($\theta = 170°$, $q = 2.5.10^7$ m$^{-1}$). As we showed above, at this q-value, the scattering intensity was very close to that of seawater itself, and it was not possible to characterize the particles by DLS directly in seawater at $\theta = 170°$. Therefore, Ter Halle et al. concentrated 1 L of seawater by a factor of 200 using ultrafiltration, which allowed them to detect colloidal particles in the seawater using light scattering at $\theta = 170°$, similar to those shown in Figure 3. However, these authors did not perform static light scattering measurements and were therefore not able to quantitatively estimate the concentration of colloidal particles. The authors claimed that these detected particles were nanoplastics using gas chromatography combined with mass spectroscopy after pyrolysis. However, with static light scattering measurements, we show that the concentration of total colloids between diameters 0.2 and 0.8 μm is not more than 1 μg/L. This extremely low concentration makes it difficult to identify the colloids with the method used by Ter Halle et al. [13].

It is likely that the colloids that were detected in the seawater samples studied here were not all nanoplastics. Therefore, we need to consider how light scattering results depend on the type of material. The radius of gyration and the hydrodynamic radius do not depend on the material. However, the light scattering intensity of particles with a given size and at a given weight concentration is proportional to their density and the square of their refractive index increment. Mineral particles are denser and have a larger refractive index increment [42]. Therefore, the estimated particle concentration would be even lower if it were assumed that they consisted of minerals instead of plastic.

The present study confirms that the detection and identification of nanoplastics in the environment is a very challenging research area. It would involve isolating enough colloidal particles from large quantities of seawater to allow for analysis with techniques such as Raman scattering [23] and gas chromatography combined with mass spectroscopy after pyrolysis [16]. The challenge is to remove all non-colloidal material and, at the same time, not introduce extraneous colloids during the isolation process.

## 5. Conclusions

The light scattering intensity of seawater samples taken at different spots off the coast of Western Europe and the Mediterranean filtered through 0.45 μm pore size filters was within the experimental error, the same as for pure water with sea salt added in the

same amount as in seawater. Comparison with model colloidal particles showed that the concentration of colloidal particles with diameters between 0.2 and 0.45 μm in the seawater samples was less than 1 μg/L. Colloidal particles were detected in seawater filtered through 0.8 μm pore size filters, but the concentration was at most 1 μg/L. The dynamic light scattering measured showed that the particles had a distribution of sizes with an average hydrodynamic diameter of 0.6 μm. The concentration of colloidal particles in the seawater samples is too low to be able to characterize their composition. Measurements on model colloidal particles show that the characterization of colloidal particles in seawater requires the use of state-of-the-art light scattering equipment that allows measurements as a function of the scattering wave vector.

**Author Contributions:** G.B.: Conceptualization, Investigation, Data curation, Writing—review and editing. F.L.: Writing—review and editing, Project administration, Funding acquisition, Supervision. C.C.: Conceptualization, Writing—review and editing, Supervision. T.N.: Conceptualization, Data curation, Writing—original draft, Supervision. All authors have read and agreed to the published version of the manuscript.

**Funding:** This work was funded by Ifremer (convention 17/1212947B, project MERLIN MICROPLAS-TIQUES) and by the ANR CESA (ANR-15-CE34-0006-02, NANOPLASTICS project).

**Institutional Review Board Statement:** Not applicable.

**Informed Consent Statement:** Not applicable.

**Data Availability Statement:** Data available in the manuscript.

**Acknowledgments:** François Galgani is acknowledged for providing the samples at different depths from the Mediterranean Sea.

**Conflicts of Interest:** The authors declare no conflict of interest.

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
