# Peer review of "Quantification of Very Low Concentrations of Colloids with Light Scattering Applied to Micro(Nano)Plastics in Seawater"

_2673-8929, doi:10.3390/microplastics2020016_

Round 1

Reviewer 1 Report

The aim of this manuscript by Gireeshkumar Balakrishnan et al. is to assess the techniques of SLS and DLS for their capacity to detect nanoplastics with diameters between d = 0.1 and 0.8 µm at low concentrations in seawater. The detection of micro(nano)plastics in environmental media is a very important topic. With the development of various detection methods, we can fully understand micro(nano)plastic distribution and their potential hazards on the ecology. This manuscript is well-written and engaging. However, before it can be accepted for publication in Microplastics, several issues must be addressed. My comments are as follows:

Major concerns:

1.      The authors should add 1-2 sentences at the beginning of their abstract to emphasize the importance of the study by declaring the hazards of micro(nano)plastics to the ecology and human health.

2.      As scholars have different definitions of nanoplastics on size, with Browne et al. defining nanoplastics as size 1 μm at the earliest (Integr Environ Assess Manag. 2007, 3(4):559-61. doi: 10.1002/ieam.5630030412.), while some scholars define nanoplastics as a size range of 1-100 nm (Rev Environ Contam Toxicol. 2012, 220:1-44. doi: 10.1007/978-1-4614-3414-6_1. Ecotoxicol Environ Saf. 2023,254:114745. doi: 10.1016/j.ecoenv.2023.114745.), the authors should use the term "micro(nano)plastics".

3.      Previous studies report some methods used for the detection of nanoplastics, such as asymmetric flow field-flow fractionation coupled to multi-angle light scattering, fluorescent labeled, and Raman tweezers (Anal Bioanal Chem. 2018, 410(22):5603-5615. doi: 10.1007/s00216-018-0919-8. J Xenobiot. 2019, 9(1):8147. doi: 10.4081/xeno.2019.8147. Sci Total Environ. 2019, 670:915-920. doi: 10.1016/j.scitotenv.2019.03.194. Environ Sci Technol. 2019, 53(15):9003-9013. doi: 10.1021/acs.est.9b03105.), but some issues remain to be addressed. The authors should make a discussion to clarify the difference between this study and other similar study (advantages and disadvantages).

Minor concerns:

4.      Line 8, “Static (SLS) and dynamic (DLS) light scattering” should be replaced by “Static light scattering (SLS) and dynamic (DLS) light scattering”.

5.      Line 44: It should be worth mentioning the earliest publication (Science. 2004, 304(5672):838. doi: 10.1126/science.1094559).

English Language is fine.

Reviewer 2 Report

I have read this manuscript with great interest and I am pleased to say that it is quite intriguing. However, before recommending publication, I suggest that the authors provide a more detailed explanation of how they prepared the microplastic patterns. In my experience, handling these materials can be challenging due to their tendency to stick to container walls and agglomerate, making it difficult to prepare valid solutions or dispersions for equipment calibration. Therefore, it is necessary to provide a detailed method, including photographs of the procedure.

That being said, I believe the manuscript is well-written and can be published. While the novelty of the research may not be groundbreaking, the results are still significant and may have important implications. The literature cited is comprehensive, as far as I know, and the technical content is accurate. The English usage is also correct, and while statistics were not utilized extensively, there is enough interesting work presented. The data interpretation is compelling and provides new conclusions. Additionally, the quality of figures is excellent.

Round 2

Reviewer 1 Report

The authors have addressed all my concerns, I can now recommend this paper for publication in the journal MICROPLASTICS.